# Absence of Association between Previous *Mycoplasma pneumoniae* Infection and Subsequent Myasthenia Gravis: A Nationwide Population-Based Matched Cohort Study

**DOI:** 10.3390/ijerph18147677

**Published:** 2021-07-19

**Authors:** Kuan Chen, James Cheng-Chung Wei, Hei-Tung Yip, Mei-Chia Chou, Renin Chang

**Affiliations:** 1Department of Emergency Medicine, Kaohsiung Veterans General Hospital, Kaohsiung 813414, Taiwan; gauurgr1016@gmail.com; 2Division of Allergy, Immunology and Rheumatology, Chung Shan Medical University Hospital, Taichung 40201, Taiwan; 3Institute of Medicine, Chung Shan Medical University, Taichung 40201, Taiwan; 4Graduate Institute of Integrated Medicine, China Medical University, Taichung 40402, Taiwan; 5Department of Management Office for Health Data, China Medical University Hospital, Taichung 40402, Taiwan; fionyip0i0@gmail.com; 6College of Medicine, China Medical University, Taichung 406040, Taiwan; 7Institute of Public Health (Biostatistics), National Yangming University, Taipei 11221, Taiwan; 8Department of Recreation and Sports Management, Tajen University, Pingtung County 90741, Taiwan; 9Department of Physical Medicine and Rehabilitation, Kaohsiung Veterans General Hospital, Pingtung Branch, Pingtung County 91245, Taiwan

**Keywords:** *Mycoplasma pneumoniae* infection, myasthenia gravis, population-based, association

## Abstract

*Mycoplasma pneumoniae* (*M. pneumoniae*) is not only one of the most common pathogenic bacteria for respiratory infection but also a trigger for many autoimmune diseases. Its infection process shared many similarities with the pathogenesis of myasthenia gravis (MG) at cellular and cytokine levels. Recent case reports demonstrated patients present with MG after *M. pneumoniae* infection. However, no epidemiological studies ever looked into the association between the two. Our study aimed to investigate the relationship between *M. pneumoniae* infection and subsequent development of MG. In this population-based retrospective cohort study, the risk of MG was analyzed in patients who were newly diagnosed with *M. pneumoniae* infection between 2000 and 2013. A total of 2428 *M. pneumoniae* patients were included and matched with the non-*M. pneumoniae* control cohort at a 1:4 ratio by age, sex, and index date. Cox proportional hazards regression analysis was applied to analyze the risk of MG development after adjusting for sex, age, and comorbidities, with hazard ratios and 95% confidence intervals. The incidence rates of MG in the non-*M. pneumoniae* and *M. pneumoniae* cohorts were 0.96 and 1.97 per 10,000 person-years, respectively. Another case–control study of patients with MG (*n* = 515) was conducted to analyze the impact of *M. pneumoniae* on MG occurrence as a sensitivity analysis. The analysis yielded consistent absence of a link between *M. pneumoniae* and MG. Although previous studies have reported that *M. pneumoniae* infection and MG may share associated immunologic pathways, we found no statistical significance between *M. pneumoniae* infection and subsequent development of MG in this study.

## 1. Introduction

The incidence of autoimmune diseases has been reported to be increasing worldwide. However, the triggers of most autoimmune diseases remain unclear. Among autoimmune diseases, myasthenia gravis (MG) is the most common neuromuscular junction disorder. The hallmark of this disorder is a fluctuating degree and variable combination of weakness in several muscle groups, such as ocular, bulbar, limb, and respiratory muscles. MG is a relatively uncommon disorder, with an annual incidence of approximately 7–23 new cases per million globally [1,2]. However, its incidence rate has been increasing since the 1950s [3,4]. MG is an acquired disease in which autoantibodies attack a patient’s own acetylcholine receptors (AchRs). The trigger factors for developing these autoantibodies are yet to be clearly identified. Among all the factors, some infections have been thought to play a major role in causing autoimmune diseases. However, no specific pathogen strongly linked to MG has been reported in the literature. Knowing what microorganism infection would raise the risk of developing MG allows us to manage it more aggressively in order to avoid a rare but devastating consequence.

*Mycoplasma pneumoniae* (*M. pneumoniae*) is one of the most common pathogenic bacteria of upper respiratory tract infection, acute bronchitis, and community-acquired pneumonia (CAP) worldwide. In Asia, 8–23% of the incidence of CAP is attributed to *M. pneumoniae* infection, which is the second most common bacterial pathogen and only falls behind *Streptococcus pneumoniae* [5]. Apart from respiratory symptoms, more importantly, *M. pneumoniae* infections are also associated with variable extrapulmonary manifestation, including neurological, cardiac, hematological, dermatological, and gastrointestinal symptoms. This phenomenon is quite unusual among other bacterial infections; however, the underlying pathophysiology remains controversial. One of the most common hypothesis is autoimmunity [6,7,8]. Many autoimmune diseases have been reported to follow *M. pneumoniae* infections, such as autoimmune hemolytic anemia, Guillain–Barre syndrome, and acute disseminated encephalomyelitis, etc. [9,10,11]. Recent studies have suggested that the excessive expression of T helper 17 cells (Th17) and associated cytokines, e.g., IL-17, could be one of the possible pathogenesis [12]. 

Alternatively, studies aimed at determining the pathogenesis of MG have revealed some connections between MG and Th17/Treg imbalance [13,14]. Figure 1 shows our working hypothesis. Although sharing those overlapping pathways in Th17 and IL 17 cascades, no previous studies have evaluated the epidemiological relationship between *M. pneumoniae* infection and subsequent development of MG. Few case reports have directly linked *M. pneumoniae* infection to subsequent MG development but no study with larger scale were carried out. Therefore, we conducted a nationwide matched cohort study to investigate whether *M. pneumoniae* infection would be associated with an increased risk of subsequent development of MG.

## 2. Materials and Methods

### 2.1. Data Source

The study used data from the inpatient datasets of the National Health Insurance Research Database (NHIRD). The NHIRD was established by the National Health Insurance program of Taiwan launched in 1995. Encryption and anonymity were performed to protect the privacy of beneficiaries. The claims data contain information regarding the basic demographics; disease diagnosis coding based on the International Classification of Diseases, 9th Revision, Clinical Modification (ICD-9-CM); and details of inpatient orders, inpatient admission, and discharge dates. The study was approved by the Research Ethics Committee at China Medical University and Hospital (CMUH-104-REC2-115(CR-6)).

### 2.2. Study Population and Main Outcome

The study population was randomly sampled by BNHI from the original claim data of NHIRD. There was no significant difference in the gender (*p* = 0.613) [15] and age distribution [16] between the patients in the randomly sampled data and the original NHIRD.

In Taiwan, the diagnosis of *M. pneumoniae* is based on a positive serologic test result [17], including the measurement of specific IgM in a blood sample, or a four-fold or greater increase or decrease in the value of a specific IgG. Suspected cases of infection are further confirmed by clinical examination or imaging [18]. Because of the strict review mechanism for antibiotic use [19], consensus must be reached among infectious physicians, chest physicians, and specialty-trained coders for all *M. pneumoniae* diagnoses. By adopting such well-established diagnostic criteria, doctors in Taiwan do their best to minimize misclassification.

The date of the initial diagnosis of *M. pneumoniae* infection (ICD-9-CM 483.0) was defined as the index date, and the end of follow-up was the date of diagnosis of MG (ICD-9-CM 358.0), the date of withdrawal or death, or 31 December 2013, whichever came first. The diagnosis of MG was validated with at least one admission or at least two consistent OPD visits examined by neurologists.

Subjects with missing registry claim data, such as unknown sex, inconsistent birth date, or incomplete insurance data, were excluded. Other exclusion criteria were as follows: the index year not between 2000 and 2012; MG diagnosed before the index date; age < 18 years; and follow-up duration less than 0. Subjects with *M. pneumoniae* infection were included in the *M. pneumoniae* cohort, whereas those who were never infected with *M. pneumoniae* were included in the non-*M. pneumoniae* cohort; *M. pneumoniae* patients were matched with non-*M. pneumoniae* patients at a 1:4 ratio by propensity score matching on 5-year age group, sex, index year, and baseline comorbidities. We followed inpatients with a diagnosis of *M. pneumoniae* who were discharged from the hospital for three outcomes: diagnosis of MG, termination of health insurance coverage, or end of the study (31 December 2013).

### 2.3. Control Population

Hospital control cohort was selected from among patients without a diagnosis of *M. pneumoniae* infection in the same database. For each patient with *M. pneumoniae* infection, we randomly selected four controls using the incidence density sampling method and matched them by 5-year age group, sex, and index year. We calculated the propensity score for admission to hospital with *M. pneumoniae* infection in the control cohort matched by age, sex, and relevant comorbidities. We defined propensity score as the probability of admission to hospital for MG conditional on baseline covariates derived by the logistic regression model (Table 1). We performed a 1:4 propensity score matching (PSM) using the greedy matching algorithm to ensure a matched pair with similar distribution of relevant comorbidities at baseline. The groups were by nearest-neighbor PSM, initially to the 8th digit and then to the first digit; that is, matching was initially done in a caliper width of 0.0000001, which was then increased for unmatched cases to 0.1. We reconsidered the matching criteria and performed a rematch (using the greedy matching algorithm). For each individual in the *M. pneumoniae* cohort, the corresponding comparisons were selected based on the nearest propensity score.

### 2.4. Covariates and Comorbidities

The baseline comorbidities included hypertension (ICD-9-CM: 401–405), diabetes (ICD-9-CM: 250), hyperlipidemia (ICD-9-CM: 272), coronary artery disease (ICD-9-CM: 410–414), chronic kidney disease (CKD) (ICD-9-CM: 585), chronic liver diseases (ICD-9-CM: 571), cerebrovascular accident (ICD-9-CM: 430–438), chronic obstructive pulmonary disease (COPD) (ICD-9-CM: 490–496), Hashimoto thyroiditis (ICD-9-CM: 245.2), Graves’ disease (ICD-9-CM: 242.0), systemic lupus erythematosus (SLE, ICD-9-CM: 710.0), rheumatoid arthritis (ICD-9-CM: 714.0), and malignancies (ICD-9-CM: 140–208). All comorbidities were recorded and traced back 2 years before the index date.

### 2.5. Statistical Analysis

To account for the difference of demographic variables and comorbidities between control and study groups, we performed a propensity score matching. We also included the covariates into the multivariable model for the adjustment.

Demographic differences between the *M. pneumoniae* and control patients were examined using independent *t*-tests for continuous variables and chi-square tests for categorial variables. We also calculated the standard mean difference for the measurement of effect size. The incidence rate was calculated by dividing the number of events by sum of the person-year in the follow-up period. To examine the independent association of *M. pneumoniae* with MG, a Cox proportional hazard regression analysis was conducted to estimate the hazard ratio (HR), and 95% confidence interval (CI). Variables that were statistically significant in the univariable model were further examined in the multivariable model. If no variables were found to be significant in the univariable model, then no further multivariable analysis was conducted. Covariates comprised age, sex, and baseline comorbidities listed in Table 1. Statistical significance was defined at *p*-value < 0.05. Data analyses and plotting were performed using SAS 9.4 (SAS Institute Inc., Cary, NC, USA).

### 2.6. Sensitivity Analysis

Because of the risk of potential source of bias and underestimating the number of MG recorded, sensitivity analysis was performed by a case control study design, comprising 515 patients with MG diagnosed and coded by neurologists, was conducted between 2006 and 2013. This sensitivity analysis was designed to test whether a history of *M. pneumoniae* infection is associated with new onset of MG. The history of *M. pneumoniae* infection was identified via physicians’ primary diagnosis from inpatients database before the diagnosis of MG within 24 months.

## 3. Results

The population consisted of 12,140 subjects; 2428 (20%) subjects were given the diagnosis of *M. pneumoniae* infection by a health care provider and 9712 (80%) subjects were not.

Table 1 shows the characteristics of patients with and without *M. pneumoniae* infection in this study. *M. pneumoniae* infection was more prevalent in the 20–40 age group, 43%, and females, 58%. DM, Grave’s disease, RA, and malignancies were less prevalent in the *M. pneumoniae* infection group. No significant differences were found in the other comorbidities between the groups.

Table 2 shows the findings of Cox regression analyses of each risk factor associated with MG among subjects. The incidence rates of MG in the non-*M. pneumoniae* and *M. pneumoniae* cohorts were 0.96 and 1.97 per 10,000 person-years, respectively. No significant difference was noted in the MG risk between the cohorts (HR = 2.14, 95% CI = [0.39, 11.72]). Similarly, no significance was found among the other covariates listed in Table 1, including hypertension (HR = 0.98, 95% CI = [0.11, 8.44]), diabetes mellitus (HR = 1.12, 95% CI = [0.2, 6.11]), hyperlipidemia, coronary artery disease, chronic kidney disease (HR = 1.59, 95% CI = [0.27, 7.97]), cerebrovascular accident (HR = 4.09, 95% CI = [0.74, 22.57]), COPD (HR = 1.57, 95% CI = [0.32, 7.78]), Hashimoto thyroiditis, Graves’ disease (HR = 5.36, 95% CI = [0.59, 48.97]), SLE, rheumatoid arthritis, and malignancies.

To confirm our finding, we conducted a case–control model sensitivity analysis using the same database for cross-validation. As shown in Table 3, the odds ratio for subsequent development of MG in *M. pneumoniae*-infected patients was found to be 0.67 (95% CI = [0.08, 5.50]). No significant association between *M. pneumoniae* infection and MG was noted in the sensitivity analysis.

## 4. Discussion

To the best of our knowledge, the present study is the first population-based nationwide study to use a longitudinal dataset (NHIRD) in order to assess the relationship between *M. pneumoniae* infection and the risk of developing MG. Our results demonstrated the lack of association between *M. pneumoniae* infection and the subsequent development of MG. A cross-validation analysis performed using a case–control method also revealed the same finding.

*M. pneumoniae* is the second most bacterial pathogen for CAP not only in Taiwan but also in most of the other countries worldwide [20,21]. In the US, it accounts for approximately 2–12% of adult cases and approximately 7.5% of children cases admitted for CAP [22]. In Taiwan, up to 14% of hospitalized CAP patients are attributed to *M. pneumoniae* [23], making it a major healthcare burden. However, this fact is often overlooked because of its mild symptoms.

Besides respiratory symptoms, *M. pneumoniae* infection also causes extra-pulmonary complications across multiple systems such as neurological, musculoskeletal, dermatological, etc. Some autoimmune diseases were also thought to be associated with *M. pneumoniae* infection, including rheumatoid arthritis (RA) [24], acute inflammatory demyelinating polyneuropathy (AIDP) [10], adult still disease [25], cold agglutinin hemolysis [9], and so forth.

The neurological complications of *M. pneumoniae* infection include meningoencephalitis, acute disseminated encephalomyelitis (ADEM), Guillain–Barre syndrome (GBS), and polyneuropathy, which are well documented in the literature. Autoimmunity is also one of the most prevalent theories, along with neurotoxin and direct invasion [26,27,28]. Some species in the *Mycoplasma* genus could produce neurotoxins. Nevertheless, currently, there is no evidence that *M. pneumoniae* could produce neurotoxins inside the human body. Direct invasion was reported in some CNS complications by direct culture. However, it would be less likely because MG is a neuromuscular junction disease.

A previous study on an animal model reported that the antigens of *M. pneumoniae* could induce potent immune reaction and enhanced Th17 cell and IL-17 response both in vivo and in vitro [29]. The investigators repeatedly inoculated specific-pathogen-free mice with *M. pneumoniae* intranasally. The result demonstrated that the *M. pneumoniae* antigen would stimulate the proliferation of mouse lymphocytes and cause the production of IL-17A and IL-10, thereby inducing a Th17 dominant immune reaction. The severity of disease is correlated with higher doses or more frequent inoculation.

In their study, Wang et al. included 30 children with *M. pneumoniae* pneumonia and 21 healthy controls to compare the frequency of Th17 cells [30]. They used flow cytometry to analyze Th17 cells in the peripheral blood in both groups. The results demonstrated that the percentage of Th17 cells increased in patients with *M. pneumoniae* infection. Subgroup analysis also revealed that the expression of Th17 cells was higher in patients with extrapulmonary manifestation than in those who did not have extrapulmonary presentation.

Guo et al. further analyzed the relationship between Th17 and Treg in *M. pneumoniae* infection [12] and found that the Th17/Treg ratio is significantly higher in *M. pneumoniae*-infected patients than in healthy controls. However, they did not address the extrapulmonary symptoms.

The Th17 and IL-17 cascades have also been proven to take part in developing autoimmune diseases [31]. Studies have found that IL-23, IL-17, Th17, and Treg cell pathways play important roles in chronic inflammation in the thymuses of MG patients [32,33]. Wang et al. reported that in MG patients with thymomas, the levels of Th17 cells and their associated cytokines increase in the peripheral blood, while those of Treg cells decrease [14]. Similar conditions did not exist in patients with a normal thymus or thymic hyperplasia in their study. However, they also found that the frequency of Th17 cells correlates with the concentration of AChR antibodies in all MG patients included.

To our knowledge, there has been two case report on ocular MG following *M. pneumoniae* infection [34,35]. Yiş et al. described a 3-year-old boy who presented with 10 days of sore throat and 3 days of bilateral ptosis. MG was diagnosed with negative MRI finding and positive neostigmine and ice pack test. *M. pneumoniae* infection was confirmed via serial serology tests. Aydin et al. also presented a 6-year-old girl with fever, nasal flow and obstruction, fatigue, and sudden onset of left eyelid ptosis developed 3 days after prior symptoms. *M. pneumoniae* infection was diagnosed by both serology and respiratory swab PCR. MG was diagnosed by empirical pyridostigmine bromide and the detect of Acetylcholine receptor antibody since the patient cannot cooperate on EMG.

Another study conducted by Iwasa et al. in Japan reported that variations in the anti-acetylcholine receptor antibody (AChR-Ab) titer in MG are significantly associated with the incidence rate of *M. pneumoniae* infection using time-series analysis with data from a single hospital in 6 years of follow-up period. They prospectively collected AChR-Ab from MG patients regularly and correlated it with data from 29 monitoring hospitals. Their findings showed that the seasonal change of AChR-Ab titer was significantly correlated with the epidemic patterns of *M. pneumoniae* infections, suggesting that *M. pneumoniae* may provoke an immune reaction in the host with or without symptoms [36].

Our study cohort is representative of and can be generalized to the general population of Taiwan. However, despite all the possible associations mentioned above, our result described an HR of 2.14 in patients with *M. pneumoniae* infection, without reaching statistically significance.

There are several limitations to our study that should be mentioned. First, in our data extracted from the NHIRD, the cases were identified purely by ICD-9-CM codes. No diagnostic and demographic data, such as family history, lifestyle, BMI, and medication use, could be retrieved. Laboratory reports including immunological data cannot be provided in the NHIRD and this is an inherent limitation in the study. Second, the number of MG patients included in the study was small. Given that MG is not a common disease and our study was conducted within a limited population, we tried to establish a cohort that resembles the real-world condition. Finally, the diagnosis of *M. pneumoniae* infection in Taiwan mostly relies on serology tests, which are reported to not be as sensitive as polymerase chain reaction [37]. However, because that the accessibility and affordability of the healthcare system in Taiwan is outstanding as opposed to those of most countries and the diagnosing protocol is relatively well established, we still consider our findings to be representative of the real-world conditions. Further studies with larger population and more MG cases are warranted to confirm our findings.

## 5. Conclusions

Although sharing immunological activation pathways at cellular levels and with similar seasonal trends, the present study found that *M. pneumoniae* infection was not associated with subsequent development of MG in Taiwan. However, it is a step forward. Since this observation study was performed in one relatively small country, if similar studies were to be done in the future in other countries of non-Asian origin, the results may be different.

## Figures and Tables

**Figure 1 ijerph-18-07677-f001:**
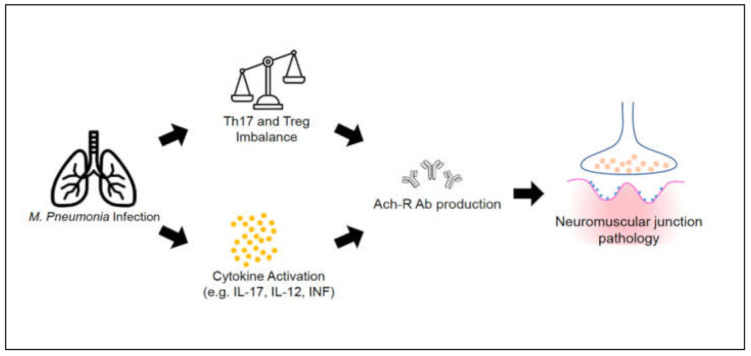
The hypothesis of the cohort study for *M. pneumoniae* infection and MG development.

**Table 1 ijerph-18-07677-t001:** The baseline characteristic and comorbidities of patients with and without *Mycoplasma pneumoniae* (*M. pneumoniae*) infection in cohort study.

	*Mycoplasma pneumoniae* Infection					
	No (*N* = 9712)		Yes (*N* = 2428)			
Variables	*n*	%	*n*	%	*p*-Value	SMD
Gender					0.013	0.057
Female	5895	60.7%	1406	57.9%		
Male	3817	39.3%	1022	42.1%		
Age, year						
20–40	3856	39.7%	1035	42.6%	0.007	0.059
40–60	3188	32.8%	794	32.7%		0.003
≥60	2668	27.5%	599	24.7%		0.064
mean, (SD)	48.7	(18.3)	47.5	(18.1)	0.002	0.069
Comorbidities						
Hypertension	1744	18.0%	403	16.6%	0.124	0.036
Diabetes mellitus	3231	33.3%	739	30.4%	0.008	0.061
-Hyperlipidemia	0	0.0%	0	0.0%		
Coronary artery disease	292	3.0%	86	3.5%	0.196	0.030
CKD	455	4.7%	102	4.2%	0.334	0.023
Chronic liver diseases	2428	25.0%	614	25.3%	0.789	0.007
Cerebrovascular accident	1401	14.4%	322	13.3%	0.151	0.034
COPD	3833	39.5%	943	38.8%	0.587	0.013
Hashimoto thyroiditis	40	0.4%	11	0.5%	0.916	0.006
Graves’ disease	212	2.2%	28	1.2%	0.001	0.080
SLE	48	0.5%	20	0.8%	0.073	0.041
Rheumatoid arthritis	19	0.2%	11	0.5%	0.040	0.045
Malignancies	833	8.6%	160	6.6%	0.002	0.075

CKD: chronic kidney disease; COPD: chronic obstructive pulmonary disease; SLE: systemic lupus erythematosus. SMD: standard mean difference (<0.1 means no difference between two group).

**Table 2 ijerph-18-07677-t002:** Incidence rate and hazard ratio of myasthenia gravis (MG).

	Myasthenia Gravis				
Variables	*n*	PY	IR	Crude HR	(95% CI)
*M. pneumoniae* infection					
No	4	41,484	0.96	1.00	(reference)
Yes	2	10,133	1.97	2.14	(0.39, 11.72)
Gender					
Female	4	31,485	1.27	1.00	(reference)
Male	2	20,132	0.99	0.81	(0.15, 4.43)
Age, year					
20–40	1	21,977	0.46	1.00	(reference)
40–60	3	17,349	1.73	3.74	(0.39, 35.96)
≥60	2	12,292	1.63	3.51	(0.32, 38.72)
Comorbidities					
Hypertension					
No	5	43,096	1.16	1.00	(reference)
Yes	1	8521	1.17	0.98	(0.11, 8.44)
Diabetes mellitus					
No	4	35,588	1.12	1.00	(reference)
Yes	2	16,029	1.25	1.12	(0.2, 6.11)
Hyperlipidemia					
No	6	50,303	1.19		
Yes	0	1315	0.00		
Coronary artery disease					
No	6	49,479	1.21		
Yes	0	2138	0.00		
CKD					
No	4	39,046	1.02	1.00	(reference)
Yes	2	12,571	1.59	1.45	(0.27, 7.97)
Cerebrovascular accident					
No	4	45,549	0.88	1.00	(reference)
Yes	2	6069	3.30	4.09	(0.74, 22.57)
COPD					
No	3	31,924	0.94	1.00	(reference)
Yes	3	19,694	1.52	1.57	(0.32, 7.78)
Hashimoto thyroiditis					
No	6	51,429	1.17		
Yes	0	188	0.00		
Graves’ disease					
No	5	50,138	1.00	1.00	(reference)
Yes	1	1479	6.76	5.36	(0.59, 48.97)
SLE					
No	6	51,384	1.17		
Yes	0	233	0.00		
Rheumatoid arthritis					
No	6	51,504	1.16		
Yes	0	113	0.00		
Malignancies					
No	6	47,644	1.26		
Yes	0	3973	0.00		

CKD: chronic kidney disease; COPD: chronic obstructive pulmonary disease; SLE: systemic lupus erythematosus; PY: person-year; IR: incidence rate per 10,000 person-year; HR: hazard ratio; CI: confidence interval.

**Table 3 ijerph-18-07677-t003:** Sensitivity analysis with case control study design.

	Myasthenia Gravis						
	No (*N* = 2060)		Yes (*N* = 515)				
Variables	*n*	%	*n*	%	OR	(95% CI)	*p*-Value
*M. pneumoniae* infection							
No	2054	99.7%	514	99.8%	1.00	(reference)	
Yes	6	0.3%	1	0.2%	0.67	(0.08, 5.50)	0.707
Gender							
Female	1163	56%	290	56%	1.00	(reference)	
Male	897	44%	225	44%	1.01	(0.83, 1.22)	0.952
Age, year							
20–40	469	23%	116	23%	1.00	(reference)	
40–60	861	42%	236	46%	1.11	(0.86, 1.42)	0.419
≥60	730	35%	163	32%	0.90	(0.69, 1.18)	0.449
mean, (SD)	54.4	(18.4)	52.5	(16.0)	0.99	(0.99, 1.00)	0.035 *
Comorbidities							
Hypertension	829	40%	199	39%	0.94	(0.77, 1.14)	0.507
Diabetes mellitus	462	22%	105	20%	0.89	(0.70, 1.12)	0.318
Hyperlipidemia	738	36%	187	36%	1.02	(0.84, 1.25)	0.837
CAD	99	5%	17	3%	0.68	(0.40, 1.14)	0.143
CKD	97	5%	12	2%	0.48	(0.26, 0.89)	0.019 *
Chronic liver diseases	586	28%	143	28%	0.97	(0.78, 1.20)	0.759
CVA	379	18%	98	19%	1.04	(0.82, 1.33)	0.742
COPD	718	35%	142	28%	0.71	(0.58, 0.88)	0.002 **
Hashimoto thyroiditis	18	1%	5	1%	1.11	(0.41, 3.01)	0.834
Graves’ disease	76	4%	20	4%	1.05	(0.64, 1.74)	0.835
SLE	28	1%	7	1%	1.00	(0.43, 2.30)	<0.99
RA	9	0.44%	2	0.39%	0.89	(0.19, 4.12)	0.880
Malignancies	253	12%	56	11%	0.87	(0.64, 1.18)	0.380

CAD: coronary artery disease; CKD: chronic kidney disease; CVA: cerebrovascular accident; COPD: chronic obstructive pulmonary disease; SLE: systemic lupus erythematosus; RA: rheumatoid arthritis; OR: odds ratio; CI: confidence interval; *: *p*-value < 0.5; **: *p*-value < 0.01.

## Data Availability

The data presented in this study are available on request from the corresponding author.

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
