# Peer review of "Absence of Association between Previous Mycoplasma pneumoniae Infection and Subsequent Myasthenia Gravis: A Nationwide Population-Based Matched Cohort Study"

_ijerph, 2021, doi:10.3390/ijerph18147677_

Round 1
Reviewer 1 Report
General comments: This is a well-written manuscript.
Introduction:
1) Need more reference and background to justify why this study is necessary. What is the working hypothesis?
2) Please consider a diagram explaining the overlap in the Th17 and IL-17 cascades of MP and MG pathophysiology that led to the hypothesis or rationalizing the study.
Methods:
1) Large sample size is a plus
2) Please comment on sample size estimation
3) Please describe any efforts to address potential sources of bias
4) How did the authors handle the missing data?
Results:
1) Need to mention the number of study participants in each group within the main text
Discussion:
1) One of the major limitations of this study is that the authors did not find any association between MG and MP. I assume that makes this a negative study. If the authors had a working hypothesis, it perhaps would have been more evident. Perhaps, it would be the responsibility of the authors to explain why this journal would publish a negative study or readers would be interested.
2) Please discuss any potential bias that the authors may have considered
Author Response
Response to reviewer:
Introduction:
1) Need more reference and background to justify why this study is necessary. What is the working hypothesis?
First of all, we thank you very much for giving us the opportunity to improve our article.
- pneumoniae has been shown to be associated with the triggering of many autoimmune diseases including autoimmune hemolytic anemia, Guillain–Barre syndrome (AIDP), CIDP, acute disseminated encephalomyelitis, asthma…etc. Case series also presented quite a few M. pneumoniae infection related neurological sequela, including MG (provided as reference #35 and #36). They drew our interests on the relationship of the two diseases. We further looked into the pathogenesis of M. pneumoniae and MG in cellular and cytokine level and found some similarity on immunological pathways (reference #12-#14 and #32-34). The other study by Iwasa, K. et al. also presented an interesting seasonal correlation with M. pneumoniae and MG. However, no study using well established database was ever carried out. That’s the starting point of our study.
2) Please consider a diagram explaining the overlap in the Th17 and IL-17 cascades of MP and MG pathophysiology that led to the hypothesis or rationalizing the study.
We added a diagram as Figure 1 in the introduction paragraph.
Methods:
1) Large sample size is a plus
2) Please comment on sample size estimation
First, we conduct this retrospective observational study by obtaining data from the database, there is no sample size estimation.
Second, the study population was randomly sampled by BNHI from the original claim data of NHIRD. There was no significant difference in the gender (P = 0.613) and age distribution between the patients in the randomly sampled data and the original NHIRD.
Please refer to Revised Version, Materials and Methods Section, Study population and main outcome Part, Page 3, Line 120-122
3) Please describe any efforts to address potential sources of bias
To account for the difference of demographic variables and comorbidities between control patients and mycoplasma pneumonia infection, we performed a propensity score matching. We also include the covariates into the multivariable model for the adjustment.
Please refer to Revised Version, Materials and Methods Section, Statistical analysis Part, Page 4, Line 221-223
4) How did the authors handle the missing data?
The NHIRD collected the data from the registry file of the insurance program and claim data, missing data rarely occurred.
Subjects with missing registry claim data, such as unknown sex, inconsistent birth date, or incomplete insurance data, were excluded
Please refer to Revised Version, Materials and Methods Section, Study population and main outcome Part, Page 3, Line 136-137
Results:
1) Need to mention the number of study participants in each group within the main text
The population consisted of 12,140 subjects. 2428(20%) subjects were given the di-agnosis of M. pneumoniae infection by a health care provider and 9712(80%) subjects were not.
Please refer to Revised Version, Results Section, Page 5, Line 263-264
Discussion:
1) One of the major limitations of this study is that the authors did not find any association between MG and MP. I assume that makes this a negative study. If the authors had a working hypothesis, it perhaps would have been more evident. Perhaps, it would be the responsibility of the authors to explain why this journal would publish a negative study or readers would be interested.
As mentioned in introduction, we started from the autoimmune diseases triggering feature of M. pneumoniae infection. We reviewed the literature and found much similarity between M. pneumoniae and MG. Epidemiological study from Japan also showed an interesting result. However, prospective study is difficult on such disease, MG, with a low incidence rate. Retrospective study should be warranted.
2) Please discuss any potential bias that the authors may have considered
All authors thank the reviewer for indicating the unclear point and giving us opportunity to clarify it.
As a retrospective cohort study, there’s some biases that’s concerning as below:
- Selection bias:
Our population was randomly selected from the original NHIRD database which consisted of more than 20 million people in Taiwan. According to the NHIRD claim, the LHID data we used could be representative of the original database
Please refer to Revised Version, Materials and Methods Section, Study population and main outcome Part, Page 3, Line 120-122
- Information bias:
The NHIRD collected the data from the registry file of the insurance program and claim data, missing data rarely occurred. - Confusion bias:
To account for the difference of demographic variables and comorbidities between control patients and mycoplasma pneumonia infection, we performed a propensity score matching. We also include the covariates into the multivariable model for the adjustment.
Please refer to Revised Version, Materials and Methods Section, Statistical analysis Part, Page 4, Line 221-223
All authors thank you for your comments.
Reviewer 2 Report
The manuscript ijerph-1212762 presents a study that demonstrated the "absence of association between previous Mycoplasma pneumonia infection and subsequent myasthenia gravis".
The study is well designed and the workflow is clear. However, the authors could grew the interest of the readers by highlighting the impact of the conclusion of the study. Why the absence of a connection between this particular microorganism and miastenia gravis is important?
The authors should specify in the abstract section why they focused on finding of a connection between M. pneumoniae and myastenia gravis. I mean, the study did not demonstrate a connection, but for clarity the objective of the study should be crystal. Maybe, adding some references about connection between autoimmune disease and M.pneumoniae. Alternatively, the references that could make a connection between the pathogenesis of myastenia gravis and M. pneumoniae.
L49 - M. pneumoniae is an important pathogen of the upper respiratory tract, but the term "pathogen" includes bacteria and viral species? I think that the sentence " is one of the most common pathogens " could be rephrased " is one of the most common pathogenic bacteria...". This is written in L183.
L60-L62 - The paragraph "he excessive production of T helper 17 cells (Th17) ..." could be rephrased. It is confusing, maybe only for me. More, the authors discussed immunological aspects but they should explain the scarcity of the immunological data from their results.
A positive aspect of the study is the study was approved by a national ethics committee.
minor comments:
the latin name of bacteria should be italic Mycoplasma pneumoniae, Streptococcus pneumoniae
In my opinion, Mp - is not a standardized abbreviation of bacterial species for scientific journals. Alternative M. pneumoniae could be used.
Author Response
Response to reviewer:
- The study is well designed and the workflow is clear. However, the authorscould grew the interest of the readers by highlighting the impact of the conclusion of the study. Why the absence of a connection between this particular microorganism and miasthenia gravis is important?
The authors should specify in the abstract section why they focused on finding of a connection between M. pneumoniae and myastenia gravis. I mean, the study did not demonstrate a connection, but for clarity the objective of the study should be crystal. Maybe, adding some references about connection between autoimmune disease and M.pneumoniae. Alternatively, the references that could make a connection between the pathogenesis of myastenia gravis and M. pneumoniae.
All authors thank the reviewer for indicating the unclear point and giving us opportunity to clarify it.
- pneumoniae has been shown to be associated with the triggering of many autoimmune diseases including autoimmune hemolytic anemia, Guillain–Barre syndrome (AIDP), CIDP, acute disseminated encephalomyelitis, asthma…etc. Case series also presented quite a few M. pneumoniae infection related neurological sequela, including MG (provided as reference #35 and #36). They drew our interests on the relationship of the two diseases. We further looked into the pathogenesis of M. pneumoniae and MG in cellular and cytokine level and found some similarity on immunological pathways (reference #12-#14 and #32-34). The other study by Iwasa, K. et al. also presented a interesting seasonal correlation with M. pneumoniae and MG. However, no study using well established database was ever carried out. We think a retrospective study is warranted to look into the association between the two.
- L49 - pneumoniae is an important pathogen of the upper respiratory tract, but the term "pathogen" includes bacteria and viral species? I think that the sentence " is one of the most common pathogens " could be rephrased " is one of the most common pathogenic bacteria...". This is written in L183.
Thank you for the correction. In the revised version, we made some correction based on your suggestion. Please refer to revised version, Introduction section, Page 2, Line 73-74
- L60-L62 - The paragraph "he excessive production of T helper 17 cells (Th17) ..." could be rephrased. It is confusing, maybe only for me. More, the authors discussed immunological aspects but they should explain the scarcity of the immunological data from their results.
We added a diagram showing our working hypothesis in the revised version. (please refer to figure 1)
Our members came into notice the interesting feature as autoimmune triggering of M. pneumoniae and found much similarity between M. pneumoniae infection and MG in the literature. Since this is a retrospective study using one of the most well-established databases that we could get, NHIRD of Taiwan, the database does not provide data on those immunological pathways. Laboratory report including immunological data cannot be provided in the NHIRD and this is an inherent limitation in the study.
Please refer to revised version, Discussion section, Page 9, Line 540-541
- A positive aspect of the study is the study was approved by a national ethics committee.
Thank you very much.
- minor comments:
the latin name of bacteria should be italic Mycoplasma pneumoniae, Streptococcus pneumoniae
In my opinion, Mp - is not a standardized abbreviation of bacterial species for scientific journals. Alternative M. pneumoniae could be used.
Thank you again for the correction. In the revised version we do fixed problem mentioned about by italicized Latin names and replaced all Mp as M. pneumoniae.
Round 2
Reviewer 1 Report
Agree with the changes
Reviewer 2 Report
The authors responded to my comments.
L244. In my opinion, "...and so forth." could be replace by "etc."